# ENSO modulates wildfire activity in China

Keyan Fang[1,2,12] ✉, Qichao Yao[3,4,12], Zhengtang Guo[5,6], Ben Zheng[7], Jianhua Du[8], Fangzhong Qi[4], Ping Yan[3], Jie Li[3], Tinghai Ou [2], Jane Liu [1,9], Maosheng He [10] & Valerie Trouet[11] ✉

China is a key region for understanding fire activity and the drivers of its variability under strict fire suppression policies. Here, we present a detailed fire occurrence dataset for China, the Wildfire Atlas of China (WFAC; 2005–2018), based on continuous monitoring from multiple satellites and calibrated against field observations. We find that wildfires across China mostly occur in the winter season from January to April and those fire occurrences generally show a decreasing trend after reaching a peak in 2007. Most wildfires (84%) occur in subtropical China, with two distinct clusters in its southwestern and southeastern parts. In southeastern China, wildfires are mainly promoted by low precipitation and high diurnal temperature ranges, the combination of which dries out plant tissue and fuel. In southwestern China, wildfires are mainly promoted by warm conditions that enhance evaporation from litter and dormant plant tissues. We further find a fire occurrence dipole between southwestern and southeastern China that is modulated by the El Niño-Southern Oscillation (ENSO).

---

[1] Key Laboratory of Humid Subtropical Eco-geographical Process (Ministry of Education), College of Geographical Sciences, Fujian Normal University, Fuzhou, China. [2] Regional Climate Group, Department of Earth Sciences, University of Gothenburg, Gothenburg, Sweden. [3] National Forestry and Grassland Administration (National Park Administration), Beijing, China. [4] China Fire and Rescue Institute, Beijing, China. [5] Excellence in Life and Paleoenvironment, Beijing, China. [6] Institute of Geology and Geophysics, Chinese Academy of Sciences, Beijing, China. [7] Department of Statistics, Colorado State University, Fort Collins, CO, USA. [8] Department of Fire Warning, Ministry of Emergency Management, Beijing, China. [9] Department of Geography and Planning, University of Toronto, Toronto, Canada. [10] Leibniz-Institute of Atmospheric Physics, Rostock, Germany. [11] Laboratory of Tree-Ring Research, University of Arizona, Tucson, AZ, USA. [12] These authors contributed equally: Keyan Fang, Qichao Yao. ✉email: kujanfang@gmail.com; trouet@arizona.edu

Wildfire is an intrinsic natural disturbance agent of vegetation and climate and is a carbon source on short timescales, but its role in the burial of charcoal and organic matter can contribute to the long-term carbon sink[1,2]. Climate determines current fuel flammability and future fuel availability and is therefore the primary large-scale driver of wildfire variability[3,4]. Under recent anthropogenically forced global warming, the frequency and severity of wildfires have increased globally, creating positive feedbacks to anthropogenic climate change on both short and long timescales[5–10]. Anthropogenic climate change can increase fire potential directly via prolonged fire seasons and hot conditions[11–13], but also indirectly by modulating oceanic and atmospheric modes, such as El Niño-Southern Oscillation (ENSO) and jet stream variability[3,14–17]. For example, the unprecedented 2019–2020 Australian wildfires that burned over ten million hectares were related to the absence of a negative (La Niña) phase of ENSO[18] (Supplementary Table 1). To draw a globally complete picture of climate-wildfire linkages, and to inform inter-governmental efforts for common actions to mitigate the global impacts of wildfires under future climate change scenarios, an improved understanding of regional wildfire responses to climate change is needed[1,2,6,19].

Temperate and boreal forest fires are modulated by high latitudinal climate modes such as the Arctic Oscillation (AO) and the polar jet stream[15,20]. Fire activity in low latitudinal regions, on the other hand, is closely linked to tropical climate drivers with regionally differing responses to ENSO (Supplementary Table 1)[14]. Wildfires in southeastern Asia and southwestern North America, for example, respond in opposite ways to ENSO[2,14]. Subtropical forests form an important and fire-prone transitional region between tropical and mid-latitudinal climate and wildfires regimes, yet there is a considerable lack of understanding about subtropical wildfire variability and its large-scale climatic drivers. The world's largest subtropical forests are located in China due to the prevailing of the Asian summer monsoon[21,22] and its mountainous terrain that is not suitable for agriculture. Chinese subtropical forests form a relatively humid "oasis" in the globally dry subtropics[23], making China a unique area to understand subtropical fire activity under a monsoonal climate.

Fire patterns in China and their linkages with climate[24–29] have mostly been studied based on provincial field survey data at coarse resolution (on average ~300,000 km²)[26,29]. Such studies have revealed broad-scale fire responses to temperature, precipitation, and wind variability, as well as oceanic and atmospheric modes, such as ENSO[24–29]. However, the climatic influences on fire activity are not well-quantified at the local scale. The high temporal and spatial resolution of individual fire occurrences needed for such local studies is only available for the country as a whole from satellite image-based fire products[27,30], which can be biased due to the interference of non-fire-related factors, such as sunlight and clouds[31,32].

Here, we present the Wildfire Atlas of China (WFAC), which is based on satellite imagery and field observations from 2005 to 2018. Most of the fires occur in southern China (SOC) and generally show a decline trend. We find a dipole fire patterns between southwestern and southeastern China (SEC) modulated by the ENSO.

## Results and discussion

### The Wildfire Atlas of China (WFAC)
The WFAC is based on the Forest Fire Prevention and Monitoring Information Center (FFPMIC) data product that combines satellite imagery and field observations and includes the location, date, and time of 135,246 fire occurrences in China (see "Methods"; Fig. 1a, Supplementary Data 1, and Supplementary Movie 1). We aggregated individual

fire occurrences into their nearest gridpoints in a $2° \times 2°$ network (number of fire occurrences per gridpoint at hourly resolution for 2005–2018) following a bilinear method to facilitate the analysis of diurnal patterns. In addition, we deleted 36 out of 219 gridpoints that had less than five fire occurrences in total (2005–2018). To study the influence of holidays and climate on fire occurrence, we then aggregated the hourly fire occurrences for each of the 183 remaining gridpoints into daily, monthly, and annual fire occurrence numbers. To develop a country-wide fire chronology from the monthly WFAC, we standardized annual fire occurrence time series for each gridpoint (to an average of zero and standardized deviation of 1) and then averaged all standardized gridpoint time series. In the same way, we developed regional WFAC fire chronologies for the ten fire regions (details below).

### Spatiotemporal wildfire characteristics
We classified the gridded monthly WFAC network into ten distinct fire regions using rotated principal component analysis (RPCA, see "Methods" and Supplemental material for details) that resemble previous spatial classifications of climate[33] and wildfire in China[24–27]. The WFAC fire chronologies for the whole country (Fig. 1a–b) and all sub-regions (Fig. 1c–f, Supplementary Fig. 1) show generally decreasing trends toward the present after reaching a peak in 2007, in contrast with the increasing fire occurrence numbers in many of the world's tropics and high latitudes[7,9,10,14]. The ten fire regions combined include 90.9% (122,952 fires) of the total number of WFAC fires and show strong interannual variability. For example, the southwestern China (SWC) fire chronology showed four times more fires in 2010 compared to the following year 2011 (Fig. 1d).

The vast majority (84%) of WFAC wildfires occurred in subtropical China (~20–30°N, 100–120°E) and this is also reflected in the results of the RPCA analysis: 90% of the fires in the ten regions (and thus more than 80% of all fires in China) occur in only four subtropical regions: SEC, SWC, SOC, and the lower reaches of Yangtze River (Fig. 1c–f). The predominance of wildfire in southern, subtropical China differs from the dominant fire patterns found in both southern and northern China in previous studies that were based on fire size rather than fire occurrence[26,27]. This combination of results suggests that fires in northern China can be of larger size to those in the south, but they occur less frequently[26,27]. The predominance of fire occurrences in subtropical China can be explained by three main factors related to fuel availability, climate, and ignition sources[34,35]. First, subtropical China is the most densely forested region in China and the world's subtropics and is characterized by high fuel availability (Fig. 1a). For example, the top ten provinces with the highest forest cover ratios in China are in the subtropics (Supplementary Fig. 2). Second, subtropical China experiences seasonal drought stress in the non-monsoon season (approximately from October to April), during which the available fuels dry out and are flammable. Third, subtropical China is densely populated and rich in human-induced ignition sources. This subtropical fire-dominated pattern in China differs from spatial fire patterns in most other regions with a low ratio of subtropical fires[32,35].

### Intra-annual and diurnal wildfire cycles
Seventy-one percent of fire occurrences in the four subtropical regions of China (68% of all ten subregions) occur in the winter season, from January to April (Fig. 2a, b). The fire season starts earliest, from January to March, in SEC (Fig. 2 and Supplementary Fig. 3), where the rainy season starts earliest in China, and fire occurrences decrease after March (Supplementary Fig. 4). In SWC, where the rainy season starts later, in May, the fire season also starts later and spans

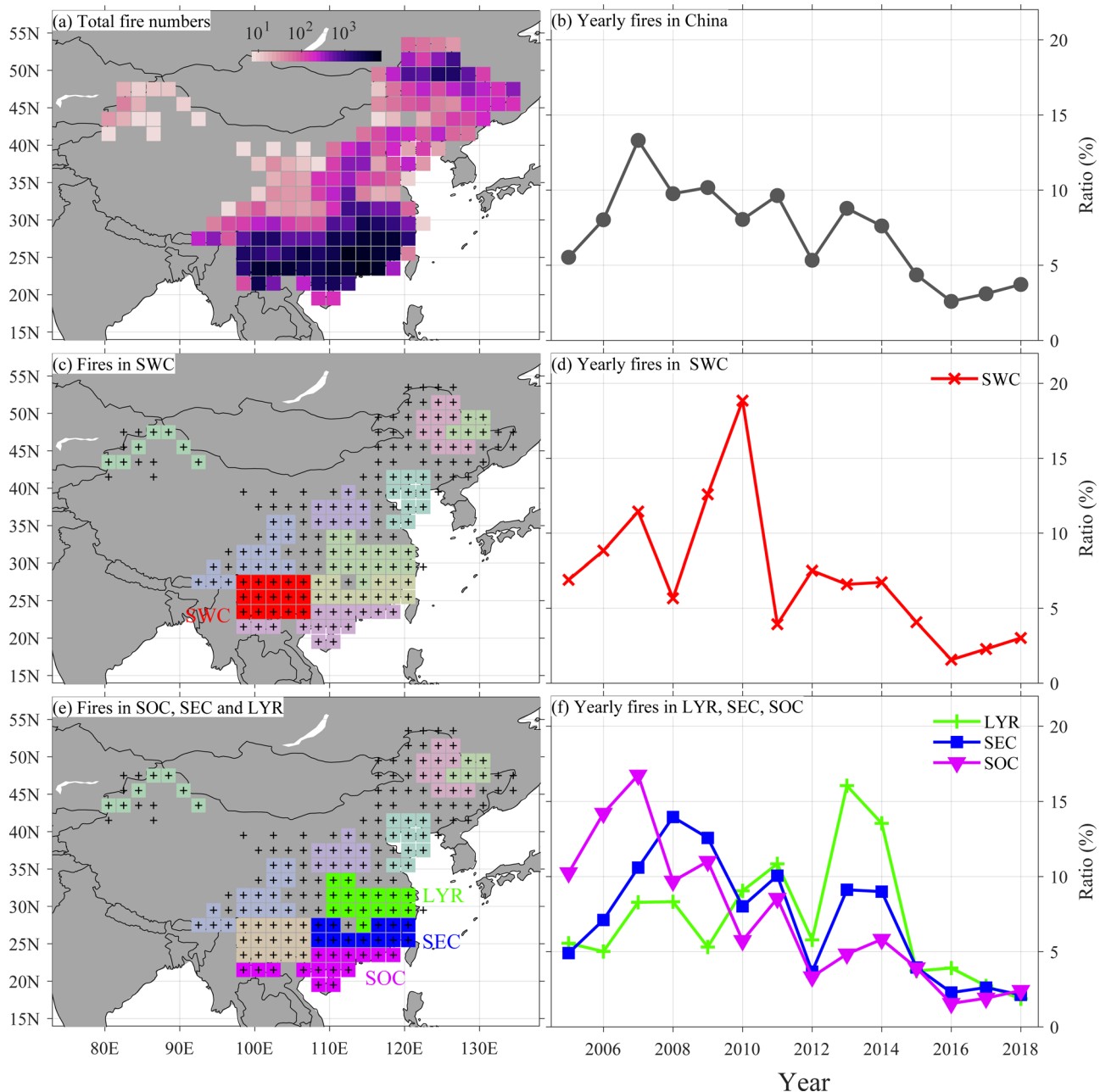

**Fig. 1 Spatiotemporal fire occurrences of the Wildfire Atlas of China (WFAC). a** Total WFAC fire occurrences (2005–2018) in the 2 × 2° gridded network, the **b** time series of the annual percentage (percentage of all fires (sum of gridpoint-level standardized fire occurrences) in a given year) of the WFAC fire chronology, the **c** location of the Southwestern China (SWC) fire region, **d** the annual percentage of fires of the SWC fire chronology, the **e** location of Southeastern China (SEC), the Lower reaches of the Yangtze River (LYR), and Southern China (SOC), and **f** the annual percentage of fires of the LYR, SEC, and SOC fire chronologies. The other shaded fire regions are shown in the Supplementary Fig. 1.

primarily from February to April. The fire season is delayed even more in northern China, from March to May, when the snow has largely melted, but the summer monsoon has not yet reached the north. For China as a whole, fire activity is lowest in summer (June to August), when the summer monsoon prevails and creates wet conditions. This monsoon related low number of fire occurrences in summer is in contrast to the peak summer fire season in boreal forests, which is driven by warm summer temperatures[6,8,20], and the dry fall fire season in Mediterranean forests, such as in California[11,15,36]. Only in the northernmost region of China, which is one of the coldest areas and at the limits of the summer monsoon, does the peak fire season occur in summer and fall (Fig. 2).

Most wildfires occur during the daytime (Fig. 2c, d and Supplementary Fig. 5) when most human-induced ignition activities take place (e.g., agricultural burning). Vegetation and fuels are also relatively dry during the daytime due to moisture loss from transpiration. Sixty seven percent of the fires occur from 14:00 to 18:00, with fire occurrences peaking roughly 2 h earlier in northern China (12:00–17:00) compared to SOC (14:00–19:00) (Fig. 2c). In dry northern China, relative humidity tends to be low enough and vegetation tends to be dry enough to catch fire earlier in the daytime than in SOC, where relative humidity often is not low enough until later in the afternoon. Earlier peak fire activity in northern China, however, may also be related to lifestyle differences in northern versus SOC. For

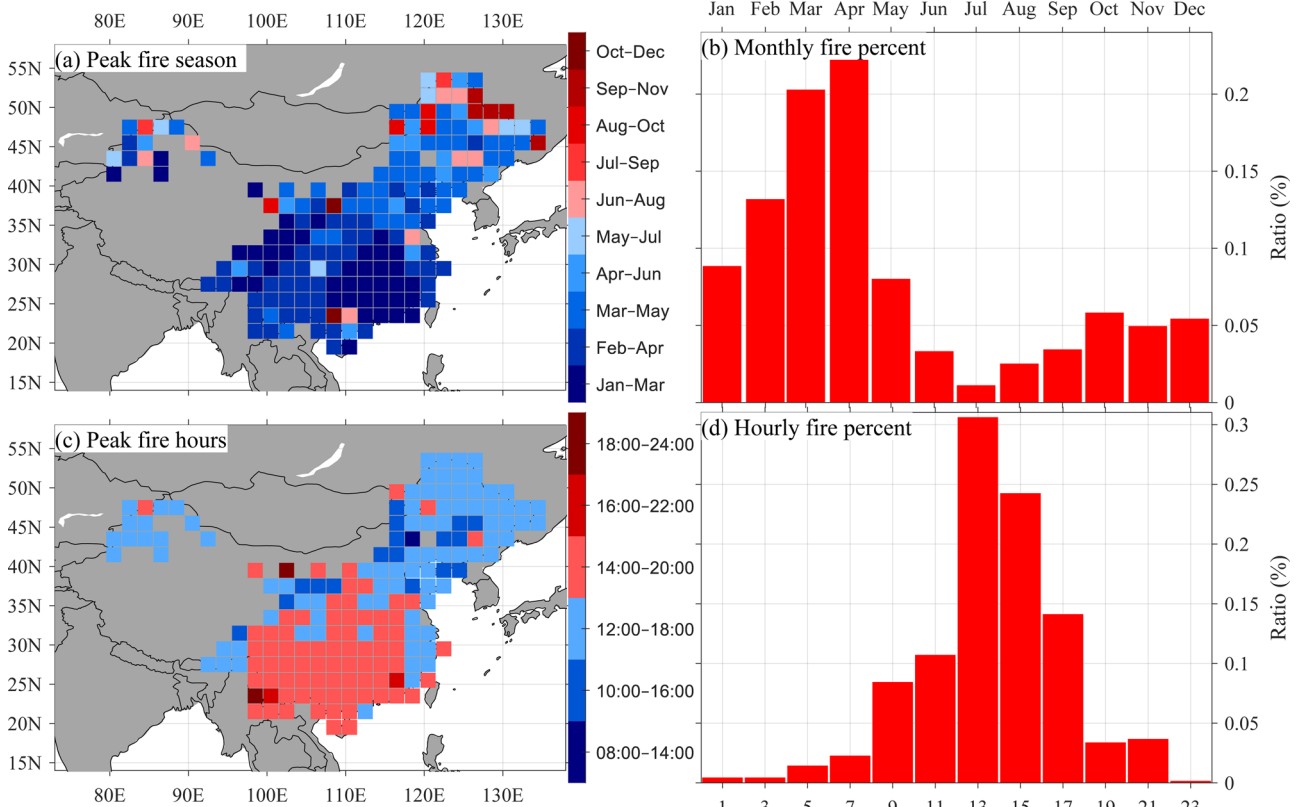

**Fig. 2 Intra-annual and diurnal fire occurrences of the Wildfire Atlas of China (WFAC). a** The peak fire season for each gridpoint containing more than 50% (mean of 73%) of the fire occurrences, **b** the monthly ratio (percentage of total fire occurrences in a given month) of the WFAC fire chronology, **c** the peak fire hours for each gridpoint containing more than 50% (mean of 77.4%) of the fire occurrences, and **d** the two-hourly ratio (percentage of total fire occurrences in a given 2-h time period) of the WFAC fire chronology.

example, due to the colder climate in the north, people generally go to sleep earlier and tend to wake up earlier, resulting in earlier ignition activities relative to the south. In warmer SOC, people tend to nap in the early afternoon, which may limit ignition and thus fire activities from 12:00 to 14:00. A clear jump in fire occurrence after 10:00 suggests that vegetation flammability increases soon after this time of day, which supports the "10 o'clock" fire prevention policy that requires all fires to be put out by 10:00 in the morning after discovery[37]. Fire activity increases most sharply after 14:00, especially in SOC, and we suggest paying special attention to this time of day for fire prevention.

In Europe and North America, fire occurrences are typically low during the weekend due to religious services and thus reduced outdoor activities[38]. This is not the case, however, in China. On the contrary, weekend fire occurrences are high in the Muslim region of northwestern China, where Friday, rather than Saturday and Sunday, is the religious service day[38] (Supplementary Fig. 6). There is also no clear increase in fire activity in northwestern China on traditional Chinese holidays. For China as a whole, however, fires occur 5.7, 5.8, and 7.3 times more frequently on traditional Chinese holidays (Chinese New Year's Eve (~February), Chinese New Year's Day (~February), and Ching Ming Holiday (~April)) than the daily mean fires of the respective holiday months (Fig. 3). The fire numbers are particularly high in north central China on Chinese New Year's Eve and in SOC on Chinese New Year's Day, which may be related to a regional difference in the day that firework is typically lighted to scare away bad fortune. Fire activity on Ching Ming Holiday is especially high in eastern China (Fig. 3), reflecting a particularly longstanding tradition to burn paper money for dead relatives in that region. In contrast, the Torch Festival (~July) for

the Yi and other minorities in SWC occurs in the low-fire monsoon season and has little impact on regional fire occurrence (Supplementary Fig. 7).

**Fire–climate relationships**. We analyzed the influence of climate on fire occurrence across the WFAC for the primary fire season (January to April), the monsoon season (May to September), the post-monsoon season (October to December), and the whole year. Fire–climate relationships are stronger for the first-differenced data than the original data (Fig. 4 and Supplementary Fig. 8), suggesting stronger fire–climate relationships on interannual than longer timescales. This may be because longer-term fire trends are related to fire suppression[26]. We found strong positive correlations between fire occurrence and fire season temperature in western and northern China (Fig. 4a and Supplementary Fig. 9). In these regions with limited fire season precipitation, high temperatures enhance evaporation, dry the fuel, and thus lead to more fire activity[26].

In SEC, on the other hand, fire occurrences are negatively correlated with fire season temperature (Fig. 4a) and precipitation (Fig. 4c), but strongly positively correlated with diurnal temperature ranges (DTR) (Fig. 4b). Fire occurrences further increase with warmer maximum temperatures in the fire season (Supplementary Fig. 9a), but decrease with warmer minimum temperature (Supplementary Fig. 9b) and increased cloud cover (Supplementary Fig. 9c). DTR and cloud cover are significantly anti-correlated in the fire season in SEC (Supplementary Fig. 10). Positive fire-DTR and maximum temperature and negative fire-cloud cover relationships indicate more fire occurrences on sunny days. High fire occurrence on sunny days may be due to intense

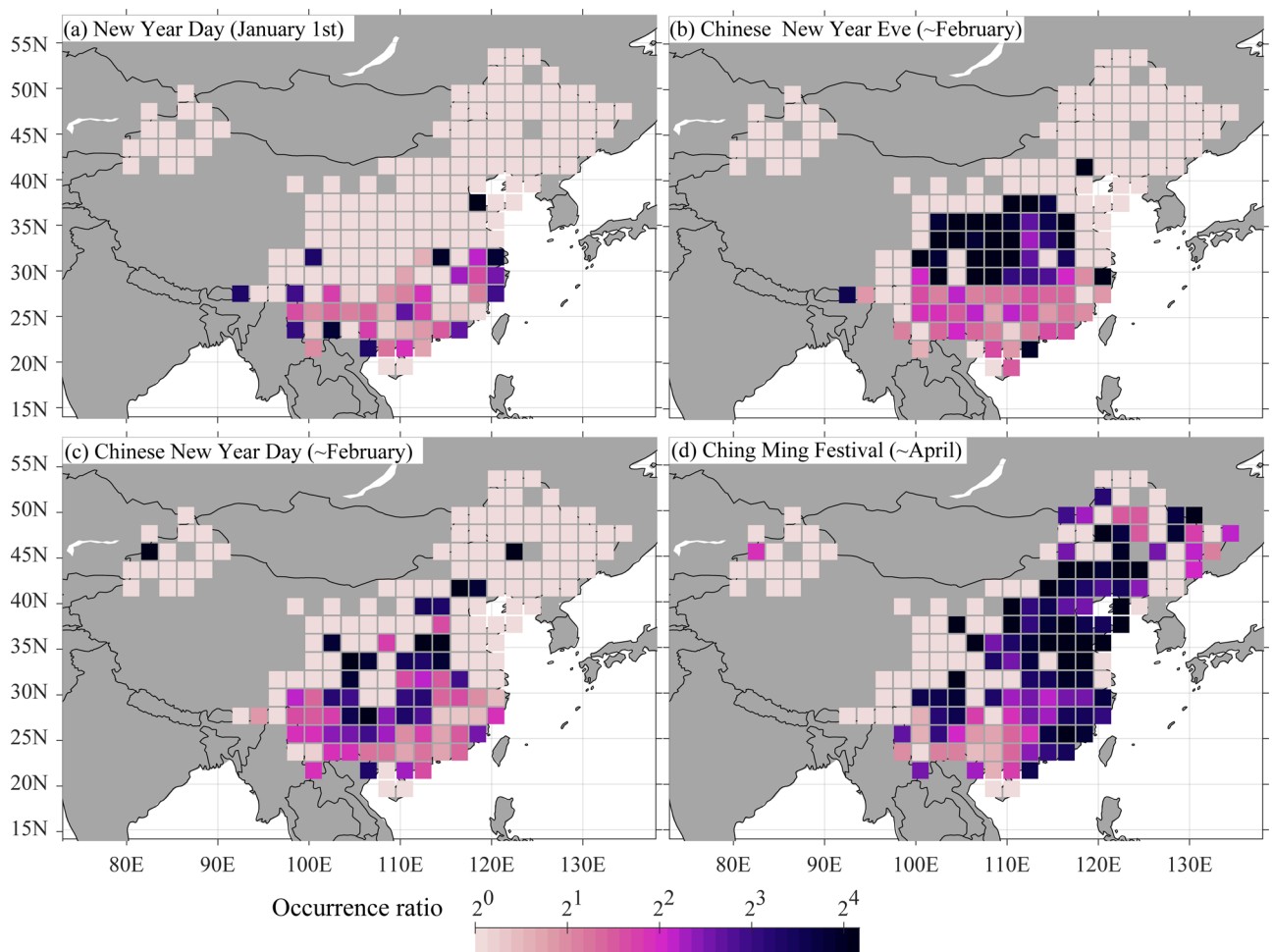

**Fig. 3 Fire occurrence ratios between the holidays and the 2005–2018 average.** The ratio between the fire occurrences on holidays of **a** the New Year Day (January 1st), **b** Chinese New Year Eve (a day before the Chinese New Year, based on the lunar calendar, often in February), **c** Chinese New Year Day, and **d** Ching Ming Festival (based on the lunisolar calendar, in April) and the mean daily fire occurrences (2005–2018) in the month of the respective holiday.

solar irradiation that can enhance evapotranspiration and moisture loss[39]. Precipitation is relatively abundant from January to April in SEC and a lack of precipitation, rather than warm temperature, plays a more limiting role on fire activity relative to mean temperature in this region.

The influence on fire occurrence of low precipitation and high DTRs in SEC and of warming in western and northern China point to drought-prone fire regimes, which is confirmed by generally negative correlations with the fire season Palmer Drought Severity Index (PDSI; Fig. 4d) and the Standard Precipitation-Evapotranspiration Index (SPEI; Supplementary Fig. 9d). Fire–climate relationships in China thus reflect fire–climate relationships in other drought-prone fire activities across the globe[1,2,11]. Fire–climate relationships are generally similar between the fire season and the entire year (Fig. 4 and Supplementary Fig. 11), largely due to the majority of yearly fire occurrences in the fire season.

Fire–climate relationships are generally weak in the monsoon season (May to September) (Supplementary Figs. 12–15), particularly in SWC with a humid monsoon climate. Monsoon-season fire–climate relationships are stronger in SEC and northern China, where the monsoon season is relatively dry compared to SWC (Supplementary Figs. 12–15). Fire–climate relationships increase in strength in the post-monsoon season (Supplementary Figs. 16–19), when fire occurrences increase with warmer

temperatures in the cold regions of central and northern China. In SEC, post-monsoon season fire occurrences, like fire season occurrences, are strongly influenced by DTR and cloud fraction.

In addition to strong regional fire–climate relationships, we found a dipole pattern between wildfire occurrences in SEC and SWC, mainly modulated by the ENSO system (Fig. 5). The dipole pattern is indicated by the first singular value decomposition (SVD) mode between the first-differenced, fire season WFAC and global sea surface temperature (SST) fields, which accounts for 53.4% of their total covariance (Fig. 5). Results were similar for the entire year and for non-detrended time series (Supplementary Figs. 20 and 21). Positive ENSO (El Niño) phases, characterized by abnormally high SSTs in the eastern equatorial Pacific Ocean (Fig. 5b), resulted in an increase in wildfire activity in SWC, but a decrease in SEC and northern China (Fig. 5a). The reverse spatial wildfire pattern occurs during negative ENSO (La Niña) years. This ENSO-modulated fire dipole pattern is confirmed by a strong negative correlation ($r = -0.97$, $p < 0.001$) between the first fire-SST SVD mode and the Niño3.4 ENSO index. Low-fire activity in SEC and northern China during El Niño phases is also in contrast to increased fire activity in southeast Asia and Australia[2,18], but in line with fire decreases in southwestern North America and northern South America[1,3,11] (Supplementary Table 1).

During El Niño years, colder-than-normal SSTs and higher-than-normal geopotential heights (GPH) occur over the western

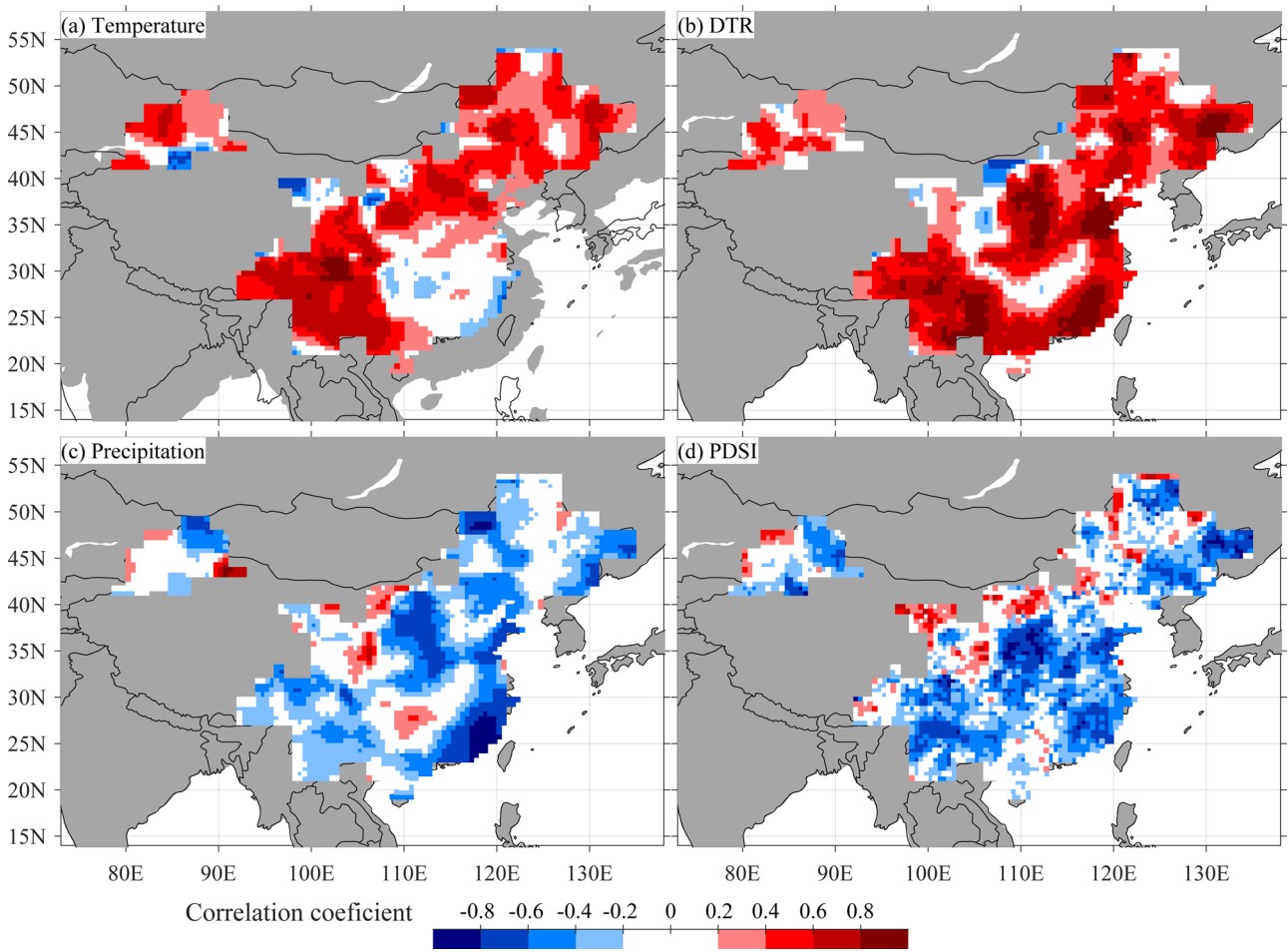

**Fig. 4 Relationships between monthly fire occurrences and climate.** Pointwise correlations between the first-differenced Wildfire Atlas of China (WFAC) fire occurrences (2005–2018) during the fire season (January to April) and average **a** temperature, **b** diurnal temperature range (DTR), **c** precipitation, and **d** Palmer Drought Severity Index (PDSI) for the same season.

Pacific and Indian Oceans[40,41], which is evidenced by positive correlations between the leading SVD pattern and 850-hPa GPH fields (Fig. 5c). A high GPH over the Bay of Bengal and associated weak India–Burma trough can weaken the southerly moisture transport to SWC and thus result in dry and warm conditions and increased fire activity[42]. This relationship is supported by the lack of significant correlations of the first SVD pattern with southerly winds over the Bay of Bengal in the fire season (Fig. 5d).

For SEC, however, the land-ocean GPH gradients with the western North Pacific Ocean increase during El Niño years (Fig. 5c, d), leading to enhanced southerly winds and strong spring persistent rain[43], the dominant precipitation pattern in China during the main fire season (Supplementary Fig. 22). The increase of precipitation can thus reduce fire occurrence in this region. Furthermore, the first SVD pattern is negatively correlated with GPH over the mid-latitudes and positively with GPH over the Arctic (Fig. 5c), reflecting a negative phase of the AO[44]. A negative AO can be associated with an enhanced Asian winter monsoon[44], resulting in cold and wet conditions and thus low-fire occurrence in northern China (Supplementary Fig. 22).

In this work, we present the first high-resolution fire occurrence dataset for China that is based on synchronous monitoring from satellite and field observations. We find a strong influence of climatic variability on wildfire occurrence in China, even though China's fire activity is strongly influenced by human activities and fire suppression[26,30]. A strict fire control policy was

implemented with almost no prescribed burning after the black dragon fire in 1987, which burned more than 10,100 km² of wildland and forest and caused the death of 213 people[26]. After this fire, fire occurrences have decreased sharply in China due to strict fire control[26,30]. The strict fire suppression policy may contribute to the decreasing fire activity after 2007. The strong association we find between fire occurrence and ENSO can inform policy-makers in considering ENSO variability in their projections of wildfire activity and in developing region-specific fire control policies.

## Methods

**Fire data**. In 2004, the FFPMIC that launched a polar orbit satellite wildfire monitoring system to ensure all vegetation in China is monitored continuously. The point location of potential fires on satellite images is confirmed by local fire prevention departments, who respond to the monitoring center within 24 h after field checking. False fires, identified in the satellite images but not observed in the field, are eliminated from the final data product. More detailed information is provided in the Supplementary material. Because of frequent missing data in the dataset for 2004, we use the FFPMIC data product from 2005 onwards. Derived from the FFPMIC dataset, the ground-truthed WFAC dataset of the number of fire occurrences differs from other satellite-based fire products of fire counts, such as the Moderate Resolution Imaging Spectrometer Terra and Aqua datasets[32]. One WFAC fire occurrence can reflect multiple satellite-based fire counts.

**Analytical methods**. We classified WFAC gridpoints into ten wildfire regions using a RPCA that maximizes the spread of individual loadings[45]. We excluded gridpoints that had the highest loadings over a specific RPCA factor, but that

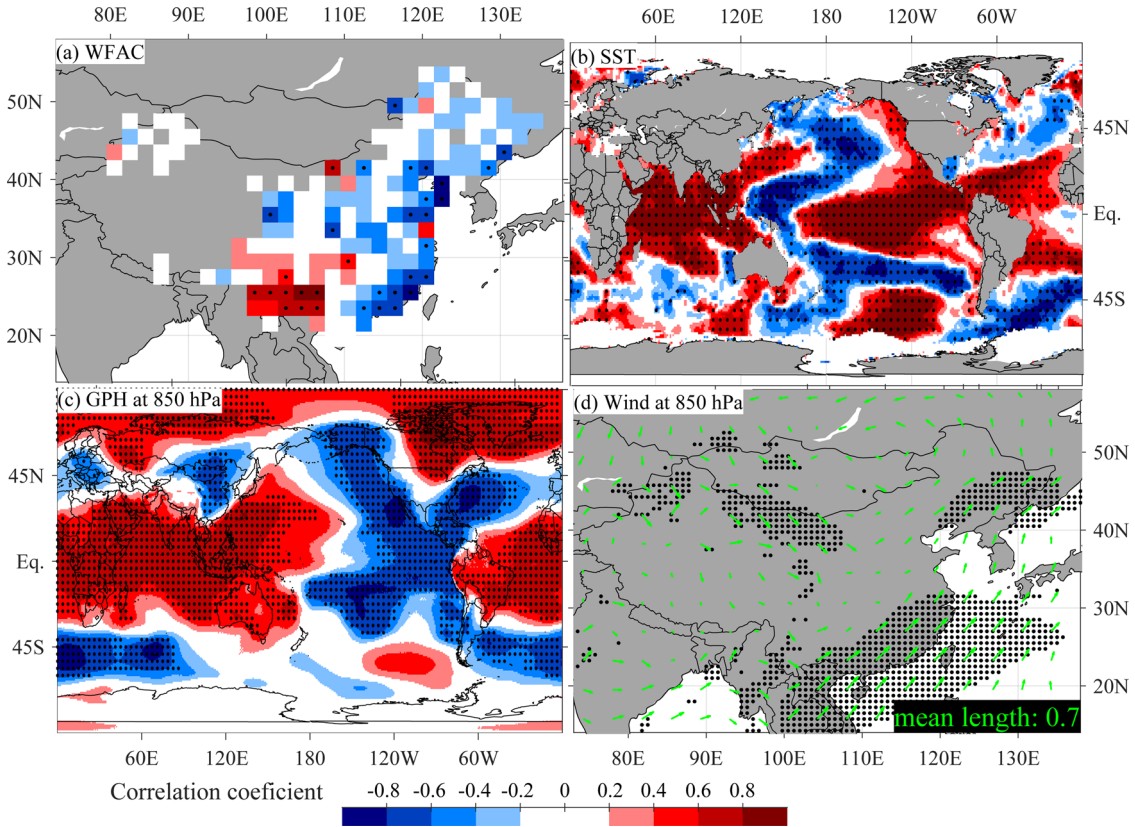

**Fig. 5 Relationships between the leading coupled fire and oceanic and atmospheric patterns.** Homogeneous correlation maps of the first leading singular value decomposition (SVD) between Wildfire Atlas of China (WFAC) and global sea surface temperature (SST) for the **a** WFAC and **b** SST. Correlations of the left singular vector (WFAC) of the first leading SVD mode with **c** the geopotential heights (GPH) and **d** the horizontal winds at 850 hPa. The SVD and correlations analyses are calculated between the first-differenced data during the main fire season January to April. Black dots denote that the significance level is above $\alpha = 0.05$. In **d**, the arrows' projections on the x- and y-axes denote the correlations with zonal and meridional winds, respectively; and the black dots denote that at least one of the correlations is above the significance level.

occurred in separate regions. The gridpoints with many years with a "0" value may cause low loadings over a specific pattern. Such gridpoints were classified into a pattern if they had the second-highest loading for a given pattern and were surrounded by gridpoints with the highest loading. In addition, we used hierarchical clustering analysis based on Pearson correlation to measure the within-group distances to validate the RPCA based classification[46].

To investigate fire–climate relationships, we correlated the monthly and seasonal gridpoint fire occurrence time series with monthly precipitation, mean, maximum, minimum temperature, DTR, PDSI, and SPEI data over their period of overlap (2005–2018). The climate data were derived from the 0.5° × 0.5° gridded Climate Research Unit dataset (CRU TS4.03)[47]. To emphasize interannual variability in the time series, we calculated Pearson correlation coefficients for the original fire and climate data, as well as for the first-differenced time series by calculating the residuals between data of two successive years.

To extract the most characteristic relationships between the WFAC and global ocean–atmosphere interactions, we applied SVD[48] on the cross-covariance matrix between WFAC and SST fields derived from the 1° × 1° gridded HadISST dataset from the Met Office Hadley Centre for Climate Prediction and Research[49]. The cross-covariance was measured by the correlation matrix between all gridpoints in WFAC and the SST field. To indicate key atmospheric processes linking the fire-SST covariability, we then correlated the coefficient of the first SVD mode for the fire season with 850-hPa GPH and wind vector fields. The GPH and wind data were derived from ERA-Interim[50], the European Centre for Medium-Range Weather Forecasts reanalysis model. We also correlated the first SVD coefficient against the January to April Niño3.4 ENSO index[40].

## Data availability

The precipitation, temperature, DTR, PDSI, and SPEI data are extracted from the Climate Research Unit dataset (https://sites.uea.ac.uk/cru/data). The SST data are downloaded from the Met Office Hadley Centre (http://hadobs.metoffice.gov.uk/hadisst/). The GPH and wind data are extracted from the ERA-Interim product from the ECMWF (https://www.ecmwf.int/en/forecasts/datasets/reanalysis-datasets/era-interim). The WFAC fire dataset produced in this study is publicly available and included in the Supplementary Information.

## Code availability

The processing codes that support the findings in this study are available from the corresponding authors upon request.

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

## Acknowledgements

We acknowledge the great efforts from collaborators in forestry departments who have monitored and checked wildfires. This study was funded by the National Natural Science Foundation of China (41888101, 42122101, and 41971022), Strategic Priority Research Program of the Chinese Academy of Sciences (XDB26020000), the State Administration of Foreign Experts Affairs of China (GS20190157002), fellowship for the National Youth Talent Support Program of China (Ten Thousand People Plan). Support from the Swedish Formas (Future Research Leaders) project is also acknowledged.

## Author contributions

K.F., Q.Y., V.T., and Z.G. designed and drafted the manuscript. K.F., Q.Y., B.Z., T.O., and M.H. conducted the analyses. V.T., K.F., Q.Y., Z.G., J.D., F.Q., P.Y., J.L., and J. Liu. commented and revised the manuscript. K.F. and Q.Y. contributed equally to the manuscript.

## Funding

## Competing interests

The authors declare no competing interests.
