## [Peer Review File · Nature Communications]

Reviewer Comments First Round -

Reviewer #1 (Remarks to the Author):

Fire is an integral component in china's terrestrial ecosystems, but is not well understood, largely due to data constraint, To address this knowledge gap, Fang et al used wildfire atlas of China and studied the spatiotemporal pattern of fire activity and its climatic drivers in China. I found this study interesting and important as it advanced our understanding on fire activity in china, and how it may change in response to climatic variations. Such knowledge is also a key component in understanding global fire activity. I would like to ask authors to make a few clarifications before I can be published.

1. L82-96. unfair comparison. MODIS active fire product detects locations of thermal anomaly at the time of satellite overpass. Highly likely that multiple thermal anomaly will be detected within the same fire as I can spread over many days.
2. L101-103: dense vegetation itself won't cause fires. I would also mention ignition source, because most fires are human-ignited in the region.
3. L206: "led to"?
4. L260: may consider change "fire regime" into fire regime, because a few key components of fire regime (e.g., fire size and severity) were not investigated.

Reviewer #2 (Remarks to the Author):

The manuscript by Fang et al. presents a comprehensive analysis of the Wildfire Atlas of China (WFAC) from 2005 to 2018. Compared to the more widely available global wildfire products such as Collection 6 of Moderate Resolution Imaging Spectroradiometer (MODIS) fire products that often overestimate the number of wildfire events, the WFAC derived from Chinese official wildfire records provides a probably more realistic depiction of the modern fire regime of China as these fire records are based both the satellite imagery and field verification. Therefore, the dataset presented in this manuscript is a valuable addition to the global fire pattern research. The authors identified a number of interesting spatiotemporal patterns of wildfire occurrence in China based on their analyses of this dataset, which include: (1) most wildfires occur in southern China of subtropical forest, with two distinct spatial clusters in southwestern and southeastern regions; (2) there is a dipole pattern of wildfire frequency in subtropical China that is modulated by ENSO, with wildfire activity being enhanced in southwestern China during El Niño years but decreased in southeastern China; (3) culture plays an important role in regulating wildfire events, with wildfire frequency spiking in traditional Chinese holidays such as Ching Ming Holiday (~ April). Such cultural influences also vary by regions partially due to the interaction of broad-scale climate, religion, and human activities. These findings are original and of extreme importance to the study of global fire patterns. The analyses are scientifically sound, and the writing is generally clear, but more analyses and clarification are needed before its publication. My major and specific comments are provided below.

(1) Most analyses of WFAC is based on a response variable called fire number, but the authors never provide a formal definition. IN my opinion, this fire number is fire frequency, which is a more widely used concept in wildfire research for describing fire regime characteristics. Fire frequency has inherent spatial and temporal dimensions. My understanding is that fire number in this research refers to the number of wildfire events at a 2 by 2-degree level over the period of 2005 – 2018 (Figure 1a). The authors further explored the monthly, daily, and hourly composition of these fire numbers in the subsequent analyses. I suggest the authors provide a formal definition of fire number and clarify its spatial and temporal dimensions.

(2) It is unclear whether WFAC was derived from the point data of fire occurrence or polygon data of fire patch. If WFAC is indeed derived from the point data, then its comparison with MODIS fire records (Figure S1) should be interpreted with caution. The high ratio of MODIS vs. WFAC in northern China is not only due to the agriculture burning, but also attributed to the fact that a single large-size fire event (e.g., 1987 Black Dragon Fire) in northern China in the WFAC dataset

may be counted as multiple fire events on separate 1-km pixels and multiple dates in the MODIS dataset. Further clarification is needed.

(3) Spatiotemporal patterns of wildfire occurrence identified in this manuscript are clear and interesting. But more analyses are needed to break down wildfire events based on the size. I suggest the authors conduct additional analyses to examine how wildfire events of large size may possess similar or different patterns of wildfires of small size.

(4) Specific comments:

a. L68-69. The authors stated a fact that there is a lack of understanding of wildfire activities in subtropical region but fails to provide a contextual background about why we should care about subtropical wildfires. Please consider adding one sentence to describe the importance of subtropical forest at the global scale.

b. L77-78. Does "Large scale" mean coarse resolution? What "detailed" do you mean here?

c. L86. Do you mean that the WFAC includes 135,246 wildfire events? You can also use wildfire occurrences in this context.

d. L87. This is the first time you use fire numbers in this paper. Please provide a formal definition here. Also, please refer to my major comment (2) regarding the comparison of fire numbers derived from WFAC and fire numbers derived from MODIS products.

e. L95. How did you standardize the number of fire occurrences? This information is crucial and should be included in the main text.

f. L179. What specific long-term trends were removed? Why would you remove the long-term trend when exploring fire-climate relationships?

g. L245. The data is not presented in the supplementary of the submission, only statistical descriptors. Would the dataset be available along with the publication of this paper?

h. L270. Do you mean the location of fire origin (point data) or the location of fire patch (polygon data)?

i. L316. Fist -> First

j. Figure 1b, 1d, and 1f. The label in y axis should be the fraction (ratio) not percent. Alternatively, multiply y value by 100 instead. This inconsistency is also a problem in many other figures.

k. Figure 5. In the caption the authors stated that the regions, where the correlation is significant at the 0.05 level, are illustrated by black dots. But I cannot find any such black dots. This is also the case in Figure S13.

l. Figure S8. The title of panel b should be DTR, to be consistent with other figures.

m. Figure S9. What is the first differenced number? And what is its advantage over the original fire number?

Jian Yang, PhD

Reviewer #3 (Remarks to the Author):

Relative to many regions of the world, the wildfires in China is relatively poor understood largely due to the absence of a robust dataset. This study presents a first detailed fire dataset for China. This dataset is highly useful for future fires studies. This study found a dipole fire pattern between southwestern and southeastern China, which is modulated by the ENSO. Their findings contribute new knowledge to advance the understanding of global wildfires, particularly for the world's subtropics. In general, this study provides key fire dataset and presents novel and robust findings. I suggest publication after addressing my comments below.

Major comments

1. The authors showed the climate-growth correlations during the pre-monsoon seasons from January to April. I suggest add further correlation analyses on the monsoon season from May to September and post-monsoon season from October to December.

2. The authors investigated the fires in some important holidays. However, one important holiday, the Torch Festival, in southwestern China was neglected. During this festival the Yi minority people often light fires, which cause wildfires. Please add analyses on the wildfires in Torch Festival.

Minor comments

1. The study compared the holiday wildfire with the daily mean wildfires for a whole year. This can lead to biases as the month with the holiday fire can be largely different from the other month. For example, the New Year Festival wildfires are low in northern China as it is not fire season in northern China in January. Comparing the holiday wildfires with the daily mean fires of a month including the holiday is suggested.

2. It is correct that the key signal is detected in subtropical China. But the analyses are always about the whole country and the ENSO showed clear influences on fire of the whole country. So I suggest to change "subtropical China" in the title to be "China".

3. It stated that a high DTR indicates a cloud free condition. Can author provide evidences to support the statement by calculating the correlations between DTR and cloud cover? Please show a correlation map about this.

4. The author found an ENSO modulation on wildfires in subtropical China. The ENSO modulations on wildfires have been widely reported for other parts of the world. It is helpful to provide a table to summarize the information on regions with strong fire-ENSO association. Such a table can be useful for us to have a global picture of the responses of fires to ENSO.

5. A brief introduction regarding the climate-fire relationships in China in previous studies should be provided.

6. "The reverse happens during La Nina years." It should be La Niña. Please correct.

7. The author said that the results can shed lights on the fire policy in China. More discussion is suggested to be provided.

Gaainho PDF Factory
www.gaainho.com

REVIEWER COMMENTS

Reviewer #1 (Remarks to the Author):

Fire is an integral component in china's terrestrial ecosystems, but is not well understood, largely due to data constraint, To address this knowledge gap, Fang et al used wildfire atlas of China and studied the spatiotemporal pattern of fire activity and its climatic drivers in China. I found this study interesting and important as it advanced our understanding on fire activity in china, and how it may change in response to climatic variations. Such knowledge is also a key component in understanding global fire activity. I would like to ask authors to make a few clarifications before I can be published.

Thank you very much for the constructive comments. We have carefully revised the manuscript to address all the comments as indicated below.

1. L82-96. unfair comparison. MODIS active fire product detects locations of thermal anomaly at the time of satellite overpass. Highly likely that multiple thermal anomaly will be detected within the same fire as I can spread over many days.

Thank you for this comment. We agree with the reviewer and have now removed this direct comparison between WFAC and MODIS datasets from the text. Instead, we indicate the differences between our dataset and the MODIS dataset in the Methods (Line 301-305).

2. L101-103: dense vegetation itself won't cause fires. I would also mention ignition source, because most fires are human-ignited in the region.

We agree with the reviewer and have largely revised this part. The revised version includes a discussion of ignition sources of subtropical fires as well as vegetation-fire relationships (Line 130-138).

3. L206: "led to"?

We revised this sentence as: "Fire-climate relationships in China thus reflect fire-climate relationships in other drought-prone fire activities across the globe^{1, 2, 11}." (Line 227-228)

4. L260: may consider change "fire regime" into fire regime, because a few key components of fire regime (e.g., fire size and severity) were not investigated.

We have changed "fire regime" in to "fire activity" in the revision (e.g., Line 31, Line

66, Line 77, Line 83, Line 151).

Reviewer #2 (Remarks to the Author):

The manuscript by Fang et al. presents a comprehensive analysis of the Wildfire Atlas of China (WFAC) from 2005 to 2018. Compared to the more widely available global wildfire products such as Collection 6 of Moderate Resolution Imaging Spectroradiometer (MODIS) fire products that often overestimate the number of wildfire events, the WFAC derived from Chinese official wildfire records provides a probably more realistic depiction of the modern fire regime of China as these fire records are based both the satellite imagery and field verification. Therefore, the dataset presented in this manuscript is a valuable addition to the global fire pattern research. The authors identified a number of interesting spatiotemporal patterns of wildfire occurrence in China based on their analyses of this dataset, which include: (1) most wildfires occur in southern China of subtropical forest, with two distinct spatial clusters in southwestern and southeastern regions; (2) there is a dipole pattern of wildfire frequency in subtropical China that is modulated by ENSO, with wildfire activity being enhanced in southwestern China during El Niño years but decreased in southeastern China; (3) culture plays an important role in regulating wildfire events, with wildfire frequency spiking in traditional Chinese holidays such as Ching Ming Holiday (~ April). Such cultural influences also vary by regions partially due to the interaction of broad-scale climate, religion, and human activities. These findings are original and of extreme importance to the study of global fire patterns. The analyses are scientifically sound, and the writing is generally clear, but more analyses and clarification are needed before its publication. My major and specific comments are provided below.

The constructive comments of the reviewer are highly appreciated. We have carefully revised the manuscript to address all of these comments.

(1) Most analyses of WFAC is based on a response variable called fire number, but the authors never provide a formal definition. IN my opinion, this fire number is fire frequency, which is a more widely used concept in wildfire research for describing fire regime characteristics. Fire frequency has inherent spatial and temporal dimensions. My understanding is that fire number in this research refers to the number of wildfire events at a 2 by 2-degree level over the period of 2005 – 2018 (Figure 1a). The authors further explored the monthly, daily, and hourly composition of these fire numbers in the subsequent analyses. I suggest the authors provide a formal definition of fire number and clarify its spatial and temporal dimensions.

We thank the reviewer for this helpful comment and now clearly define what we mean by fire number. The fire number we used concerns the number of wildfire occurrences and as such we now refer to it as 'fire occurrences' rather than 'fire number'. We also

provide the necessary details about its spatial and temporal dimensions. The original FFPMIC fire monitoring dataset includes location, date, and time of each fire occurrence. We aggregated hourly fire occurrences per day of the year for each 2° grid cell for the investigations of diurnal and holiday fire patterns. We additionally aggregated the fire data into WFAC with a monthly resolution to facilitate monthly fire-climate correlation analyses (Line 95-102).

(2) It is unclear whether WFAC was derived from the point data of fire occurrence or polygon data of fire patch. If WFAC is indeed derived from the point data, then its comparison with MODIS fire records (Figure S1) should be interpreted with caution. The high ratio of MODIS vs. WFAC in northern China is not only due to the agriculture burning, but also attributed to the fact that a single large-size fire event (e.g., 1987 Black Dragon Fire) in northern China in the WFAC dataset may be counted as multiple fire events on separate 1-km pixels and multiple dates in the MODIS dataset. Further clarification is needed.

We agree with this important comment and have now removed the direct comparison between WFAC and MODIS datasets from the text. Our WFAC data represents the number of fire occurrences per grid cell, and thus point data of fire occurrence rather than polygon data. One long-lasting fire may refer to many fire counts in the MODIS product, but only one fire occurrence in WFAC (Line 301-305).

(3) Spatiotemporal patterns of wildfire occurrence identified in this manuscript are clear and interesting. But more analyses are needed to break down wildfire events based on the size. I suggest the authors conduct additional analyses to examine how wildfire events of large size may possess similar or different patterns of wildfires of small size.

We agree with the reviewer that fire sizes are important to consider, but unfortunately, the WFAC dataset is based on point data, not burn area and thus does not reflect fire size. To emphasize the importance of fire size, we have added discussion on the spatial patterns of wildfires by referring to previous fire studies (Line 125-129) of different fire sizes. Previous studies revealed that fires occur more frequently in southern China, yet that fire sizes are higher in northern China. Thus, our fire occurrence-based study indicates dominant fire patterns in southern China, whereas fire size-based studies show dominant patterns over both southern and northern China.

(4) Specific comments:

a. L68-69. The authors stated a fact that there is a lack of understanding of wildfire activities in subtropical region but fails to provide a contextual background about why we should care about subtropical wildfires. Please consider adding one sentence to describe the importance of subtropical forest at the global scale.

Thanks to the reviewer's comments, we have added a sentence in the introduction to

clarify the importance of wildfire in subtropical forests, as they grow in transitional zones from fires regimes under tropical climate drivers to mid- and high latitude climate conditions (Line 70-73).

b. L77-78. Does “Large scale” mean coarse resolution? What “detailed” do you mean here?

We have revised the text to avoid misunderstandings. By 'large scale' we indeed mean coarse resolution and by 'detailed' we mean records reflecting individual fire occurrences (Line 79-84).

c. L86. Do you mean that the WFAC includes 135,246 wildfire events? You can also use wildfire occurrences in this context.

Yes, it includes 135,246 wildfire occurrences. We revised the text to use wildfire occurrence (Line 92-95).

d. L87. This is the first time you use fire numbers in this paper. Please provide a formal definition here. Also, please refer to my major comment (2) regarding the comparison of fire numbers derived from WFAC and fire numbers derived from MODIS products.

Fire number in our manuscript refers to the number of fire occurrences. Thanks to the reviewer’s comments, we have now changed the term 'fire number' to 'fire occurrence' and have added a definition of the term (Line 92-102; Line 301-305). We realized the problems of direct comparisons between fire counts in MODIS products and the WFAC and have now removed this comparison from the manuscript.

e. L95. How did you standardize the number of fire occurrences? This information is crucial and should be included in the main text.

We have now added information about our standardization methods to the main text (Line 102-106). We standardized the fire occurrence time series for each grid cell to have an average value of 0 and standard deviation of 1. We then averaged fire occurrence time series for all grid cells to produce the fire chronology for China as a whole, as well as associated regions.

f. L179. What specific long-term trends were removed? Why would you remove the long-term trend when exploring fire-climate relationships?

We now explain the specifics of our detrending methods, as well as the reason for doing so in the revised manuscript (Line 200-204; Line 321-324). We detrended the fire occurrence time series because they show strong trends that are most likely related to fire suppression. To explore fire-climate relationships, such anthropogenic

fire trends need to be removed. Nevertheless, all correlation coefficients were calculated based on the raw (undetrended) and detrended (for both fire and climate) time series.

g. L245. The data is not presented in the supplementary of the submission, only statistical descriptors. Would the dataset be available along with the publication of this paper?

We have a data availability statement in the revised submission and will share the dataset produced in this study (WFAC) after acceptance of the manuscript.

h. L270. Do you mean the location of fire origin (point data) or the location of fire patch (polygon data)?

Fire occurrences in our study refer to fire origin (point data), not fire polygons. We have now revised the text to clearly define this (Line 92-102) and avoid misunderstandings.

i. L316. Fist -> First

Corrected in the revision.

j. Figure 1b, 1d, and 1f. The label in y axis should be the fraction (ratio) not percent. Alternatively, multiply y value by 100 instead. This inconsistency is also a problem in many other figures.

We have revised all the related figures (Fig. 1 and supplementary Fig. S1) with the y axis as ratio and multiplied by 100 according to the reviewer's comments.

k. Figure 5. In the caption the authors stated that the regions, where the correlation is significant at the 0.05 level, are illustrated by black dots. But I cannot find any such black dots. This is also the case in Figure S13.

This is a very valid comment, and we have now increased the size of the black dots in the revised Figs. 5 and supplementary Fig. S20-22.

l. Figure S8. The title of panel b should be DTR, to be consistent with other figures.

m. Figure S9. What is the first differenced number? And what is its advantage over the original fire number?

We have revised the title of Figure S8b as DTR. We have added explanation on the motivation, calculation method, and advantages of the first differenced data in the revision (see also earlier comment; Line 200-204; Line 321-324).

Jian Yang, PhD

Reviewer #3 (Remarks to the Author):

Relative to many regions of the world, the wildfires in China is relatively poor understood largely due to the absence of a robust dataset. This study presents a first detailed fire dataset for China. This dataset is highly useful for future fires studies. This study found a dipole fire pattern between southwestern and southeastern China, which is modulated by the ENSO. Their findings contribute new knowledge to advance the understanding of global wildfires, particularly for the world's subtropics. In general, this study provides key fire dataset and presents novel and robust findings. I suggest publication after addressing my comments below.

We thank the reviewer for their constructive comments. We have carefully revised the manuscript to address all comments below including 10 additional figures.

Major comments

1. The authors showed the climate-growth correlations during the pre-monsoon seasons from January to April. I suggest add further correlation analyses on the monsoon season from May to September and post-monsoon season from October to December.

We agree with this excellent suggestion of the reviewer and have calculated climate-fire correlations for the monsoon and post-monsoon seasons. In general, the fire-climate relationships weaken in the monsoon season, probably due to its humid climate, and become strong again in the post-monsoon season with the decay of monsoon and thus dry climate. To present the results of these analyses, we have added 8 figures to the supplemental information (supplementary Fig. S12-19) and discuss the results in the main text (Line 233-241).

2. The authors investigated the fires in some important holidays. However, one important holiday, the Torch Festival, in southwestern China was neglected. During this festival the Yi minority people often light fires, which cause wildfires. Please add analyses on the wildfires in Torch Festival.

The reviewer makes a very valid point. We have now added a figure (Fig. S7) and description (Line 193-195) of the fire occurrences related to the Torch festival. The Torch festival occurs during the low-fire season in July and August with humid monsoonal climate. Therefore, we find that the Torch festival has limited impact on fire occurrences.

Minor comments

1. The study compared the holiday wildfire with the daily mean wildfires for a whole year. This can lead to biases as the month with the holiday fire can be largely different from the other month. For example, the New Year Festival wildfires are low in northern China as it is not fire season in northern China in January. Comparing the holiday wildfires with the daily mean fires of a month including the holiday is suggested.

Thank you for this excellent comment. We now compare fire occurrences on holidays to the daily mean fires of the month with holidays, as the reviewer suggests, and have revised Fig. 3 to present these new results.

2. It is correct that the key signal is detected in subtropical China. But the analyses are always about the whole country and the ENSO showed clear influences on fire of the whole country. So I suggest to change “subtropical China” in the title to be “China”.

Revised accordingly.

3. It stated that a high DTR indicates a cloud free condition. Can author provide evidences to support the statement by calculating the correlations between DTR and cloud cover? Please show a correlation map about this.

Based on the reviewer’s very helpful comment, we have added a correlation map to illustrate the relationship between DTR and cloud cover (Fig. S10). We found significantly negative correlations between DTR and cloud cover, particularly in southern China where most wildfires occur (Line 214-217).

4. The author found an ENSO modulation on wildfires in subtropical China. The ENSO modulations on wildfires have been widely reported for other parts of the world. It is helpful to provide a tale to summarize the information on regions with strong fire-ENSO association. Such a table can be useful for us to have a global picture of the responses of fires to ENSO.

We have now added a table (Table S1) that lists global fire-ENSO relationships and discuss this table in the main text (Line 57-59, Line 66-70, Line 255-258).

5. A brief introduction regarding the climate-fire relationships in China in previous studies should be provided.

We now summarize the current knowledge about climate-fire relationships in China based on previous studies in the introduction (Line 81-84).

6. “The reverse happens during La Nina years.” It should be La Niña. Please correct.

Corrected accordingly.

7. The author said that the results can shed lights on the fire policy in China. More discussion is suggested to be provided.

We thank the reviewer for their excellent suggestion. We have now suggested fire policy in China to be based on fire-ENSO linkages at the end of the text (The strong association we find between fire occurrence and ENSO can inform policy-makers in considering ENSO variability in their projections of wildfire activity and in developing region-specific fire control policies.) (Line 281-291).

Reviewer Comments Second Round -

Reviewer #1 (Remarks to the Author):

I have read the revision and authors' response to comments. The authors have addressed my concerns and I have no further comments.

Reviewer #2 (Remarks to the Author):

This revision almost fully addressed the concerns I had in the previous round of review. I only have two minor comments. 1) please consider change "of equal size to" to "larger size than" in L131 as fires in northern China tends to be large in size. 2) please add detailed information about the Wildfire Atlas of China (WFAC) dataset. Specifically, describe the meaning of each column in the supplementary excel table.

Reviewer #3 (Remarks to the Author):

I have checked the revised manuscript and found that the authors have taken effort to solve my concerns. The manuscript has been improved greatly. The present paper conducted additional studies during the monsoon and post-monsoon seasons and showed reasonable results. This study revealed key linkages between fire and ENSO, which will contribute for developing region specific fire policies. In general, I am satisfied with the revision and suggest the manuscript is acceptable in its current form.

Gaaino PDB Triad
www.gaaino.com

REVIEWERS' COMMENTS

Reviewer #2 (Remarks to the Author):

This revision almost fully addressed the concerns I had in the previous round of review. I only have two minor comments. 1) please consider change "of equal size to" to "larger size than" in L131 as fires in northern China tends to be large in size. 2) please add detailed information about the Wildfire Atlas of China (WFAC) dataset. Specifically, describe the meaning of each column in the supplementary excel table.

Thank you for the thoughtful comments. We have corrected the sentence in L131 to use large size according to the reviewer's comments. We added more detailed information (a paragraph of readme information) to the WFAC excel table, including the meaning of each column. We additionally carefully checked and edited the manuscript.